# A Cross-Sectional Client Satisfaction Study Among Persons Living with HIV Attending a Large HIV Treatment Centre in Trinidad

**DOI:** 10.3390/healthcare13121400

**Published:** 2025-06-12

**Authors:** Jonathan Edwards, Sharon Soyer, Noreen Jack, Gregory Boyce, Verolyn Ayoung, Selena Todd, Robert Jeffrey Edwards

**Affiliations:** Medical Research Foundation of Trinidad and Tobago, Port of Spain 150123, Trinidad and Tobago; jonathan.r.a.edwards@gmail.com (J.E.); ssoyer@mrftt.org (S.S.); gboyce@mrftt.org (G.B.); stodd@mrftt.org (S.T.)

**Keywords:** satisfaction, HIV, viral suppression, confidentiality, wait time

## Abstract

**Background:** Client satisfaction with HIV service delivery reflects the ability of healthcare providers to effectively deliver care and treatment that meets the requirements and expectations of clients, and is associated with improved health outcomes, including increased retention in care and HIV viral suppression. The aim of the study was to conduct a client satisfaction study among PLHIV attending a large HIV clinic in Trinidad to identify the gaps in service delivery and factors associated with reduced HIV viral suppression. **Methods**: This cross-sectional study was conducted over the period April 2023–March 2024 among 362 clients attending the HIV clinic. A structured, pre-tested questionnaire collected demographic data and factors that affected client clinic experiences, including wait time, communication with staff, confidentiality, physical amenities and HIV viral suppression. Multivariable logistic regression was used to assess the likelihood of reporting satisfaction based on key independent variables. **Results:** Among participants, 219 (60.5%) were females, 202 (55.8%) were aged 30–49 years and 337 (93.1%) were virally suppressed. Participants reported satisfaction with overall care (95.3%), confidentiality (95.9%) and interactions with doctors (96.1%), nurses (98.6%) and other staff. Dissatisfaction was reported with facility-related, elements including the outdoor/tented waiting area (46.1%), the toilet/washrooms (37.0%) and the clinic wait time (31.8%). Participants were less likely to be satisfied with the amount of medication received if they had unsuppressed viral loads (*p* = 0.035), were aged 20–29 years old (*p* = 0.048) or had a tertiary education (*p* = 0.008). **Conclusions:** The study showed that 93.1% of the study participants were virally suppressed, and there was a general level of satisfaction with the overall care at the clinic, confidentiality and healthcare workers’ service delivery; however, gaps involving the physical facilities, wait times and medication services should be prioritized.

## 1. Introduction

The HIV and AIDS pandemic is entering its 6th decade, with significant strides in the bio-medical treatment of the virus. The creation of an enabling environment has resulted in persons living with HIV (PLHIV) reclaiming their lives, and therefore becoming productive members of society.

In Trinidad and Tobago (T&T), it is estimated that there are approximately 10,000 persons currently living with HIV/AIDS, and approximately 70% are on antiretroviral therapy (ART) [1]. Effective HIV treatment results in HIV viral suppression, averts the sexual transmission of HIV, and results in patients living longer and healthier lives [2]. Suboptimal retention in care and poor adherence to ART are the major barriers to effective HIV viral suppression [3], and clients who are satisfied with the clinical services offered have an increased retention in HIV care [4,5,6].

Client satisfaction with HIV service delivery is associated with the perceived standard of care [7] and it reflects the ability of healthcare providers to effectively deliver care and treatment that meets the requirements and expectations of clients [8]. Patient satisfaction may be evaluated from the characteristics of the provider–patient relationship, including the adequacy of staffing, privacy, confidentiality and communication with healthcare workers; the services offered and wait time at the clinic; and the health facility environment, including location, accessibility, physical amenities, sanitation and hygiene [8]. Studies conducted in Cameroon [9], Ethiopia [8,10,11] and Nigeria [12] have shown that patient satisfaction may be influenced by sociodemographic factors, including age, sex, educational level, occupation and marital status [8,10]. Client dissatisfaction with clinical services may result from ineffective communication, long wait times, disrespect and perceived rudeness by the healthcare providers, and poor health literacy [13]. Clients who are dissatisfied with health services are less likely to adhere to their treatment regimens, have reduced health service utilization and increased clinic default, may experience worse clinical outcomes and may have a tendency to get into potential disagreements with their providers [13].

Trinidad and Tobago (T&T) comprises the southernmost islands of the Caribbean archipelago, with a population of approximately 1,368,333 persons [14], and encompasses a single nation. T&T is a multi-ethnic and multicultural nation, comprising 35.4% persons of East Indian descent, 34.2% persons of African origin, 23.0% mixed-race persons and 7.4% persons of other ethnicities [15]. The first cases of HIV/AIDS were diagnosed in 1983 in men who have sex with men (MSM) [16], with an evolution to the heterosexual transmission of HIV in 1985 [17]. All PLHIV in T&T have free access to healthcare, including antiretroviral therapy (ART) which became available in 2002, subsidized by the government [15]. The Medical Research Foundation, Trinidad and Tobago (MRFTT) is the largest HIV clinic in T&T. It provides treatment and care to PLHIV in a centralized location in Port of Spain, Trinidad, and there are two smaller sites that offer a tailored package of services, including enhanced psychosocial support for non-virally suppressed clients. The clinic has an enrolled population of approximately 10,700 clients since the commencement of HIV treatment in 2002, and there are currently approximately 5200 patients in active care and follow up. The clinic provides differentiated care services [18], with all programs having the common goal of helping PLHIV achieve and maintain optimum health through sustained HIV viral suppression.

HIV patients experience stigma and discrimination from both the general public and healthcare workers [19,20], and this may contribute to nonadherence to ART and the non-suppression of their HIV viral loads [21,22]. Poor HIV viral suppression has serious public health and individual consequences, including increased HIV transmission in the population, potential for HIV drug resistance, increased patient morbidity and mortality, and decreased survival due to HIV-related complications [23]. Studies have shown that clients who are satisfied with the clinical services comply better with care, resulting in increased retention in care, adherence to ART and HIV viral suppression [3].

Data are sparse regarding client satisfaction with HIV treatment services in the Caribbean; thus, it is important to assess these services from patients’ perspectives in order to understand the baseline, identify areas of concern and assess interventions in healthcare provision to the identified problems, with the goal of achieving HIV viral suppression. The aim of the study was to conduct a client satisfaction survey among PLHIV attending the HIV clinic, MRFTT, to identify the gaps in service delivery at the clinic and recognize factors that may contribute to reduced HIV viral suppression, so that appropriate interventions can be put in place.

## 2. Methods

### 2.1. Population and Sample

A cross-sectional study was conducted over the period 15 April 2023–31 March 2024 at the HIV clinic, MRFTT, which involved the administration of a pre-tested questionnaire to collect basic demographic data, ethnicity, education, religion, nationality, and factors that affect client clinic experience such as wait time, interactions with staff, clinic location, cleanliness and comfort of the building, privacy/confidentiality, patient adherence to ART and HIV viral suppression. The questionnaire was adapted from a validated instrument used in the study by Buluba et al. [24], where the reliability of the tool was tested and the Cronbach’s alpha was 0.71; however, no formal validation of the questionnaire used in our study was performed. Data on HIV viral loads were extracted from the patient files.


Inclusion Criteria:
Patients 18 years and older;Enrolled at the HIV clinic, MRFTT, >12 months;On ART > 12 months;Able to give informed consent.Exclusion Criteria:
Patients under the age of 18 years;Enrolled at the HIV clinic, MRFTT, <12 months;On ART < 12 months;Patients who are not able to give informed consent.



To calculate the sample size at the 95% confidence level with a margin error (confidence interval) of 5, using a population size of 5200 at a population proportion of 50%, the calculated sample size would be 358 study patients.

### 2.2. Procedures

The questionnaire was pre-tested on 10 clients attending the HIV clinic to identify and address problems/ambiguities before the main study was started. Over the period 15 April 2023–31 March 2024, participants were recruited for the study using convenience sampling. Approximately 80–120 clients attend the MRFTT HIV clinic daily and the study was conducted about 1–3 days per week depending on the availability of the study staff. Patients were informed about the research study by a doctor or a nurse, and were asked about their willingness to participate. After obtaining written informed consent, the self-administered questionnaire was given to study participants. An average of 5–10 completed questionnaires were collected on each day the study was conducted.

### 2.3. Ethical Considerations

Ethical approval to conduct the study was obtained from the Campus Research Ethics Committee of the University of the West Indies, St Augustine, approval number CREC-SA.2073/03/2023. The questionnaire was administered by doctors and nurses at the MRFTT after written informed consent from clients was obtained. No compensation was provided for participation, and clients who refused were not denied any service or resource.

### 2.4. Data Collection Instrument

A structured pre-tested questionnaire was used to assess client satisfaction, which participants completed independently. The instrument comprised both closed-ended and Likert-scale questions, covering a broad range of domains, including greeting by staff, privacy/confidentiality, respectfulness of interactions, wait times, clinic cleanliness, and the effectiveness of communication with healthcare providers. Questions were categorized into two main rating scales: (1) Always, Sometimes, and Never for service quality; and (2) Excellent, Good, Fair, Poor, and Terrible for facility reviews.

### 2.5. Variables and Definitions

The primary outcome variable was client satisfaction, measured through responses to 27 survey questions. Satisfaction levels were categorized based on the frequency of positive responses. Key demographic and clinical variables were collected categorically and included sex, age group, marital status, employment status, educational attainment, nationality, religion, and viral load status (undetectable vs. detectable). Viral suppression (undetectable HIV viral load) is defined by the World Health Organization as patients having a HIV viral load of ≤1000 copies/mL.

### 2.6. Statistical Analysis

Descriptive statistics were used to summarize demographic characteristics and satisfaction scores. Variables are presented as frequencies and percentages.

Bivariate analyses were conducted to examine associations between demographic variables and satisfaction scores. Responses categorized as Excellent and Good were combined to represent favorable satisfaction, while Fair, Poor, and Terrible were combined to represent unfavorable satisfaction. Odd ratios (ORs) with 95% confidence intervals (CIs) were calculated using logistic regression to assess the likelihood of reporting positive satisfaction outcomes based on key independent variables such as sex, viral suppression status, and age groups. Multivariable logistic regression is a suitable methodological choice for analyzing patient satisfaction/dissatisfaction based on multiple independent variables such as the quality of service, sex, viral suppression, age groups, client wait time, etc. This statistical methodology is particularly valuable because it can be used to distinguish binary outcomes (satisfaction/dissatisfaction) and complex associations among variables. Analyses were performed in RStudio 2021.09.0.

## 3. Results

During the period 15 April 2023–31 March 2024, using convenience sampling, 362 clients attending clinic appointments, were invited to participate in a Client Satisfaction Study at the MRFTT; 219 (60.5%) were females and 143 (39.5%) were males, and of these (n = 358), 306 (85.5%) self-identified as heterosexual and 52 (14.5%) self-identified as homosexual/bisexual. Among the males, 32 (8.9%) self-identified as homosexual and 16 (4.5%) as bisexual, and among the females, 1 (0.3%) self-identified as lesbian and 3 (0.8%) as bisexual.

Among the 362 clients, the ethnic distribution was as follows: 213 (58.8%) were of African descent, 37 (10.2%) East Indian, 110 (30.4%) mixed race, and 2 (0.6%) clients self-identified as other ethnicities (Table 1).

In the study, 337 (93.1%) of the study participants were virally suppressed with a viral load < 1000 copies/mL.

Of all study clients, 224 (61.9%) reported being single, 62 (17.1%) were married, 49 (13.6%) were in a common law relationship, 12 (3.3%) divorced and 15 (4.1%) widowed.

Client age was documented in ranges from <20 years to 60+ years. No study participants were <20 years of age and the majority (124 (34.1%)) of clients were aged 40–49 years, and 202 (55.8%) were aged 30–49 years.

The employment status reported by the study participants (n = 359) was as follows:202 (56.3%) employed, 39 (10.8%) self-employed, 85 (23.7%) unemployed, 31 (8.6%) retired and 2 (0.6%) students (Table 1).

### 3.1. The Physical Setting

The MRFTT is in an urban location, around a busy thoroughfare known as “the Savannah”, next door to the newly commissioned Ministry of Health headquarters and in close proximity to the General Hospital in Port of Spain. The location is easily accessible using public transportation, but the pedestrian and vehicular access faces the busy thoroughfare.

The building’s location boasts minimum signage and the clinic facilities include doctors’ rooms, cubicles for nurses and social workers, a reception area, a phlebotomy room, a laboratory that conducts HIV viral load tests, CD4 counts and other laboratory tests, and a pharmacy where patients collect the antiretroviral drugs and medications to treat opportunistic infections/sexually transmitted infections (STIs). The clients’ waiting areas comprise an indoor area closer to the doctor’s offices and an outdoor tiled area covered by a tent where patients wait for phlebotomy and to collect the medications from the pharmacy. The medication collection area is a window that opens into the outdoor tented area where ART is dispensed and advice on taking medications is provided by the pharmacists. There is one indoor toilet/washroom and one outdoor toilet/washroom which are shared by male and female clients. The building is not very big and the staff utilize the available space while being cognizant of the sensitivity of the service, so various strategies are employed to maintain privacy and confidentiality of the clients.

Participants were asked to rate their opinion of the physical amenities of the clinic on a scale ranging from Excellent to Terrible (Table 2), and favorable responses (excellent/good) were scored for the indoor waiting area (90.3%). Unfavorable responses (Fair/Poor/Terrible) were scored for the outdoor tented area (46.1%), the toilet/washrooms area (37.0%), the physical setting/location (29.8%), the medication collection area (22.7%) and the general cleanliness of the building (22.4%).

### 3.2. Interaction with Different Cadres of Staff

The MRFTT has a multi-disciplinary team, and clients interact with different cadres of staff to meet their different needs. This study explored participants’ views of these experiences/interactions with the staff who provide the most face-to-face interactions at the MRFTT and clients were asked to rate their interactions in general, and not with specific individuals (Table 2).

Participants assigned scores of Excellent/Good for their interactions with the doctors, 348 (96.1%); nurses, 357 (98.6%); phlebotomists, 353 (97.5%); receptionists, 339 (95.5%) and pharmacists, 339 (95.5%). Pharmacists are often the last and important stop where clients are given their medication, have their dosing and regimens verified, and adherence assessed.

At the MRFTT, telehealth was formalized as an intervention during the COVID-19 pandemic, and its success resulted in a continuation of the service in clients booked for appointments the same or the following week. Telehealth experiences with the nurses included medication adherence assessments, health and wellness counseling, the completion of clients’ blood investigation forms and the generation of prescriptions in readiness for when the clients physically present to the clinic. Of the 331 clients who experienced telehealth with the nurses, 315 (95.2%) reported the experience as Excellent/Good.

Clients at MRFTT are referred to social workers for psycho-social services that include general counseling, grief counseling, psycho-social needs and assessments, including financial and food insecurity needs and transportation challenges. Their role is critical in closing the gap and being proactive in preventing at-risk clients from defaulting from the clinic and non-adherence to ART. Of the 268 participants who reported interactions with social workers, 254 (94.8%) reported these interactions as Excellent/Good.

### 3.3. Perception of the Quality of Service

Using the rating scale of Always, Sometimes or Never, participants were asked to review a statement and choose the option which best described how they felt, with Always recorded as Excellent/Good and Sometimes/Never as Fair/Poor/Terrible. The aim was to get an indication of their perception of the quality of service they receive at MRFTT (Table 2).

Study participants were asked if, upon entering the compound, they felt welcomed by the staff, 319 (88.1%) clients said they Always felt welcomed whereas 41 (11.3%) said they only Sometimes felt welcomed. Only 2 (0.6%) clients said they Never felt welcomed upon entering the clinic.

Participants were asked if the staff who greeted them on entry to the clinic was respectful, and 347 (95.9%) clients reported that this was Always the case, while 15 (4.1%) reported that this was the situation only experienced Sometimes.

It is important, given the nature of the clinic, that clients’ privacy/confidentiality is maintained, and 347 (95.9%) reported that this was always the case and 4.1% reported that this was never maintained at the clinic.

Participants were also asked if they felt respected when they attended the clinic, and 351 (97.1%) reported that they were always respected by staff and 337 (93.1%) participants reported they were always treated fairly while at the clinic.

Participants were asked additional questions to ascertain their perceptions of care. Using the Likert scale ranging from Excellent, Good, Fair, Poor or Terrible, clients were asked to assign the most appropriate rating in response to the statement presented (Table 2).

Participants were asked about the waiting time at the clinic, and 247 (68.2%) of clients said that the waiting time was Excellent–Good, and 115 (31.8%) stated the waiting time was unfavorable (Fair/Poor/Terrible). All participants were asked to rate their perception of the overall quality of care they received at the clinic, and 345 (95.3%) clients reported that they found the care to be Excellent/Good, and 17 (4.7%) reported their overall care to be unfavorable (Table 2).

In Table 3, statistically significant multivariable logistic regression analysis demonstrated that participants were less likely to be satisfied with the amount of medication received if they were aged 20–29 years, (OR 0.32, 95% CI (0.12–0.86), *p* = 0.025, if they obtained a tertiary education compared to those with secondary education, OR 0.44, 95% CI (0.21–0.94), *p* = 0.034, and if they had unsuppressed HIV viral loads, OR 0.37, 95% CI (0.15–0.93), *p* = 0.035 (Table 3).

Participants were less likely to be satisfied with the toilet/washroom facilities if they were aged 20–29 years, OR 0.46, 95% CI (0.21–0.99), *p* = 0.048; aged 50–59 years, OR 0.55, 95% CI (0.31–0.98), *p* = 0.043, and if they obtained a tertiary education compared to those with a secondary education OR = 0.41, 95% CI (0.24–0.69), *p* = 0.008.

Study participants who were retired were less likely to be satisfied with the cleanliness of the building compared to those who were employed, OR 0.22, 95% CI (0.10 0.48), *p* < 0.001.

## 4. Discussion

Patient satisfaction with HIV care and treatment programs is very important as this can improve the overall quality of care and result in better patient outcomes. Patients who are satisfied with their care are less likely to default from clinic visits and more likely to adhere to their ART regimens, which leads to better health outcomes [25,26], including HIV viral suppression, reduced morbidity and mortality, improved survival and reduced transmission of HIV to sexual partners [27]. Thus it is incumbent on HIV program managers and healthcare workers to focus on the key aspects of the components of healthcare delivery and service that could impact patient satisfaction [9,28].

The MRFTT is doing admirably as a HIV care and treatment facility, and is well on its way to achieving the UNAIDS target of 95% of patients on ART being virally suppressed by 2030 [29], as 337 (93.1%) of the study participants were virally suppressed with a viral load < 1000 copies/mL. Thus, it was not surprising that when clients were asked to rate their perception of the overall quality of care received at the clinic, 345 (95.3%) clients reported that they found the care to be Excellent/Good. The high level of satisfaction with the quality of HIV care received was similar to that seen in other studies conducted in Tanzania [24], Nigeria [12] and Cameroon [9]. This is in contrast to studies conducted in Uganda [30], Vietnam [31] and Pakistan [32], where issues such as poor communication by staff, transportation costs, alcohol consumption, drug use and inadequate seating at clinics resulted in lower levels of client satisfaction. Our study findings showed that there was no association between the overall quality of care received and HIV viral suppression. In contrast, a cross-sectional study conducted among adults receiving HIV care and treatment at two clinics in Texas [3] showed that participants who achieved viral suppression were statistically significantly more satisfied with their HIV care compared to patients who did not and this was associated with increased retention in care and adherence to ART [3].

Privacy and confidentiality are important tenets of HIV clinical care and impact client satisfaction with the services received. Due to the stigma associated with HIV, a number of undesirable consequences may result if a client’s HIV status is disclosed without their consent, including discrimination, psychosocial trauma, an unwillingness to engage with medical care and social isolation [24]. Therefore, privacy and confidentiality provide a more comfortable environment, and a safe space for the client, which is essential to promoting open communication with healthcare workers, resulting in better optimal care [24,33]. In our study, 347 (95.9%) clients stated that their privacy/confidentiality was always maintained, and similar results were obtained in a study by Buluba et al. in Tanzania [24]; however, this was in contrast to a study conducted by Tran et al. in Vietnam [27], where 60.1% of clients were satisfied with the privacy and confidentiality of the healthcare services. In addition, poor physical infrastructure that results in overcrowding of the clinic and a lack of privacy may impact patient engagement and retention in care [34]. Therefore, it is recommended that capacity building programs for healthcare workers in the field of medical ethics, legal frameworks around client privacy, reporting mechanisms and consequences for breaches, should be instituted to ensure that health providers adhere to professional standards [24].

The study showed that there were areas in the physical amenities and location of the clinic that required improvement, as unfavorable responses (Fair/Poor/Terrible) were scored for the outdoor/tented waiting area (46.1%), the toilet/washroom area (37.0%), the cleanliness of the building (22.9%) and the medication collection area (22.7%). Participants were less likely to be satisfied with the toilet/washroom facilities if they were aged 20–29 years, OR 0.46, 95% CI (0.21–0.99), *p* = 0.048; aged 50–59 years, OR 0.55, 95% CI (0.31–0.98), *p* = 0.043; and if they obtained a tertiary education, OR = 0.41, 95% CI (0.24–0.69), *p* = 0.008. Clients also expressed dissatisfaction (22.9%) with the cleanliness of the building and were less likely to be satisfied if they were retired compared to those who were employed, OR 0.22, 95% CI (0.10 0.48), *p* < 0.001. Similar unfavorable responses to the physical environment were found in studies conducted in South Africa and Pakistan due to the unavailability of drinking water, location of the toilets and general cleanliness [28,32]. The HIV clinic MRFTT has since been renovated, the building has been painted, the toilets have been refurbished, and more seating has been put in the outdoor/tented area, which is now properly covered to reduce the risk of patients being affected by rain and wind.

Due to stigma and discrimination, 29.9% of the study participants expressed dissatisfaction with the location of the building as clients do not like to be seen entering the HIV clinic which is located on a busy thoroughfare, and members of the key population groups such as MSM, transgender persons and sex workers who have a higher HIV prevalence experience additional stigma and discrimination due to their sexual identities and behaviors [35]. Therefore, clients are encouraged to wear masks (as COVID-19 protocols are still in place) or they are offered the option of attending the extended hours clinic from 3 pm to 7 pm when fewer persons are around [15] or they can attend the two other MRFTT clinic sites that are in more discrete locations. Other strategies to mitigate stigma include community outreach initiatives and the integration of telehealth services.

The wait time is the time period clients enter the clinic to the time their visit has been concluded and they leave the clinic, and 31.8% of clients stated that the wait time was unfavorable. The longest wait time was for those clients (newly enrolled clients or those with unsuppressed viral loads) who have to receive more comprehensive services at their clinic visit, including physician and/or social worker visit, laboratory investigations and antiretroviral medication pick-up, which averages 45–60 min in comparison to a clinic visit of 5–10 min for medication pick-up. In a study conducted in Vietnam by Tran et al. [31], where 55.4% of clients were dissatisfied with the wait times, this may have been attributed to poor healthcare facilities and limited numbers of healthcare workers [31]. In this study, 46.1% of the study participants were injection drug users, where it was identified that there were inadequate services for these patients [31]. This is in contrast to a study conducted in Tanzania where 85.3% were satisfied with the wait time [24] possibly due to an adequate number of healthcare workers most who had a very good work ethic and for those study subjects who accessed fewer services such as medication pick up [24]. Increased wait times at HIV clinics may adversely impact HIV viral suppression by reducing patient engagement in care (as they have to take time off from work with the potential for loss of wages), medication adherence, and retention in care [36,37]. A study by Hickey et al. [36] showed that shorter wait times can improve viral suppression, especially among those who are ART-experienced [36]. However, a study by De Schacht et al. [38] showed that increased time spent at a HIV treatment facility was not associated with a lower odds of viral suppression nor a lower retention in care [38].

In our study, the nurses conducted telehealth sessions with clients which were also used as a form of triage as prescriptions were generated, blood collection forms completed and appointment dates and times given to reduce wait time in the clinic. In addition, there are television screens with educational videos and talks given by councilors that provide engaging waiting room activities [27]. Some benefits of telehealth include increased retention in care for patients who live long distances from the clinic, privacy for patients who do not want to be seen attending an HIV clinic and greater flexibility in booking appointments [39]. The limitations of telehealth include disparities in the use of technology by racial minority groups, older individuals and those with low telehealth literacy and lack of access to appropriate devices, broadband internet and appropriate policies for patients’ data protection [40]. On the day of their clinic appointment, clients are first seen by a nurse and directed to the appropriate service to improve clinic flow. Invariably on their clinic date, clients neglected their appointment time and arrived at clinic very early in the hope they will be seen quickly but this increases congestion at the clinic and prolongs the wait time.

Effective communication between clients and healthcare providers is associated with improved patient adherence and better overall quality of care [28,41]. In our study, there were reported favorable communication/interactions with doctors (96.1%), nurses (98.6%), pharmacists (95.5%), social workers (94.8%), phlebotomists (97.5%) and receptionist (95.5%), and clients reported that they always felt welcomed (88.1%) and respected (97.0%) by the staff. Training to reduce stigma and discrimination among PLHIV and effective communication/listening skills has been conducted for staff of the MRFTT, and this is reinforced during the orientation of new staff members. However, studies conducted in Cameroon [9] and KwaZulu Nata [28] reported less favorable communication/interactions between study participants and health workers who were too busy to listen to their concerns, and there was a language barrier in some situations [9,28].

The availability of ART is an important concern for clients, as stockouts may put them at increased risk of treatment discontinuation, interruption in care, treatment failure, onward HIV transmission, drug resistance, and increased morbidity and mortality [24,42]. In our study, 326 (90.1%) clients were satisfied with the amount of medication received at the clinic and ART is offered at no cost to clients [15]. Similarly in studies conducted in Tanzanian [24], Nigerian [12] and Ethiopian [43] respondents were satisfied with the accessibility of ART which is offered free of charge. However, in our study, participants were less likely to be satisfied with the amount of medication received if they were aged 20–29 years, OR 0.32, 95% CI (0.12–0.86), if they had unsuppressed HIV viral loads, OR 0.37, 95% CI (0.15–0.93), and if they obtained a tertiary education, OR 0.44, 95% CI (0.21–0.94). Similarly, lower levels of satisfaction among those with higher education levels were reported in studies conducted in four PEPFAR-supported African countries [27], Ethiopia [41] and Nigeria [44]. Patients who are more educated may be better informed and thus have higher expectations of the healthcare services than those who are less educated and possibly less informed [44]. In contrast, other studies conducted in Ethiopia reported higher levels of satisfaction among patients with higher levels of education compared to those with no formal education [10,11].

Generally, a three-month supply of ART is given to patients. Trinidad and Tobago is considered a high-income country and thus does not have access to generic dolutegravir due to patency issues [45], though tenofovir/lamivudine/dolutegravir (TLD) is used as a first-line ART regimen in other Caribbean countries [46]. Our first-line regimen is a single tablet regimen (STR) of efavirenz/tenofovir/emtricatibine; however, as most countries have transitioned to dolutegravir-based regimens, there is a shortage of efavirenz-based STR globally [47], and sometimes, patients may receive only a one month supply of the two-pill combination of efavirenz plus tenofovir/emtricitabine. This results in an increased pill burden, more frequent visits to clinics with increased transportation costs, more time off from employment, overcrowding and increased wait times, reduced adherence and unsuppressed viral loads. However, in a study conducted in Ethiopia, though antiretroviral (ARV) medication is available free of charge at HIV clinics, 46% of participants disagreed that the availability of the prescribed ARV medication was adequate, and there was decreased satisfaction among those patients who did not receive enough medication, as this resulted in incurred costs due to having to purchase ARVs from private pharmacies [48].

This study has a number of limitations. One of the study eligibility requirements was for participants to be attending the clinic for >12 months, thereby excluding newly registered clinic patients. Newly registered patients may not have formed any bonds with the healthcare workers or patterns of retention in care, and adherence to ART, and may be at a greater risk of being lost to follow up [3]; thus, it would be more challenging for them to assess the satisfaction with HIV care. This was a cross-sectional study with convenience sampling, and may thus not be generalizable to other PLHIV attending the clinic, as well as PLHIV populations in other countries of the world who may have characteristics that differ from the study population, as participants in the study were chosen on easy accessibility rather than by a random sampling method. No formal validation of the questionnaire used in our study was performed, which may present psychometric weaknesses and reduce the possibility of more accurately interpreting the results related to patient satisfaction. There may be the potential for self-report bias due to social desirability, as some participants may exaggerate their satisfaction with the clinical services due to concerns about stigma or perceived ramifications, resulting in inaccurate data collection. The predominant use of categorical variables limited the application of more robust statistical analyses, such as correlations or regressions, which would have allowed for a more detailed investigation of factors associated with satisfaction and viral suppression. It is recommended that future studies use continuous variables and a formal validation process using psychometric analyses to improve quantitative evaluations so that more detailed conclusions can be obtained. However, the findings are, in general, positive and provide guidance to the health service management team in the identification of the key factors to improve the quality of care for all patients and enhance the attainment of viral suppression. Strategies are being implemented to address the concerns of clients, including renovations of the physical infrastructure.

## 5. Conclusions

The study showed that 93.1% of the study participants were virally suppressed, there was a general level of satisfaction with the overall care at the clinic, as well as with healthcare worker services, and privacy/confidentiality was maintained. Interventions such as infrastructure enhancements, efficient scheduling of patients and streamlining workflows to reduce wait time, more efficient forecasting of medication to reduce the possibility of stock outs, educational programs focused on treatment adherence, and strategies that strengthen continuity of care will be implemented to improve service delivery, especially in patients with a higher risk of defaulting from care and those with unsuppressed viral loads. Future longitudinal studies will be implemented to evaluate the effectiveness of these interventions using more advanced methodological approaches to strengthen public health policies.

## Figures and Tables

**Table 1 healthcare-13-01400-t001:** Baseline characteristics of the study population.

Descriptive Statistic	Total (n = 362)
**Age group (years)** **20–29** **30–39** **40–49** **50–59** **60+**	34 (9.4%) 78 (21.5%) 124 (34.3%) 79 (21.8%) 47 (13.0%)
**Sex** Males Females	143 (39.5%) 219 (60.5%)
**Ethnicity** African East Indian Mixed race Other	213 (58.8%) 37 (10.2%) 110 (30.4%) 2 (0.6%)
**Education** None Primary school Secondary school Vocational Tertiary/University	3 (0.8%) 68 (18.8%) 174 (48.1%) 26 (7.2%) 91 (25.1%)
**Employment (n = 359)** Employed Self-employed Unemployed Retired Student	202 (56.3%) 39 (10.8%) 85 (23.7%) 31 (8.6%) 2 (0.6%)
**Sexual Orientation (n = 358)** Heterosexual Gay/Lesbian/Bisexual	306 (85.5%) 52 (14.5%)
**Marital status** Single Married Common law Divorced Widowed	224 (61.9%) 62 (17.1%) 49 (13.6%) 12 (3.3%) 15 (4.1%)
**Virally suppressed (Viral load < 1000 copies/mL)** Yes No	337 (93.1%) 25 (6.9%)

**Table 2 healthcare-13-01400-t002:** Client assessment of the amenities at the clinic and interactions with Staff.

	Excellent/Good	Fair/Poor/Terrible
Physical setting/location	254 (70.2%)	108 (29.8%)
General cleanliness	281 (77.6%)	81 (22.4%)
Waiting area Indoor area Outdoor area/Tent	327 (90.3%) 195 (53.9%)	35 (9.7%) 167 (46.1%)
Medication collection area	280 (77.3%)	82 (22.7%)
Washroom area	228 (63.0%)	134 (37.0%)
Communication with staff Doctors Nurses Pharmacists (n = 355) Social workers (n = 268) Phlebotomists Receptionists (n = 355)	348 (96.1%) 357 (98.6%) 339 (95.5%) 254 (94.8%) 353 (97.5%) 339 (95.5%)	14 (3.9%) 5 (1.4%) 16 (4.5%) 14 (5.2%) 9 (2.5%) 16 (4.5%)
Feel welcomed by staff	319 (88.1%)	43 (11.9%)
Feel respected by staff	351 (97.0%)	11 (3.0%)
Privacy/Confidentiality is maintained	347 (95.9%)	15 (4.1%)
Waiting time at clinic	247 (68.2%)	115 (31.8%)
Amount of medication received	326 (90.1%)	36 (9.9%)
Overall quality of care at the clinic	345 (95.3%)	17 (4.7%)

**Table 3 healthcare-13-01400-t003:** Significant predictors of patient satisfaction: results from multivariable logistic regression models by satisfaction domain.

Satisfaction Outcome Domain	Key Independent Variable	Multivariable Analysis *^,†^	
		**OR (95% CI)**	***p* value**
**Amount of medication received**	Age: 20–29 years	0.32 (0.12–0.86)	0.025
	Level of Education:Tertiary Education	0.44 (0.21–0.94)	0.034
	Viral Load: Unsuppressed (>1000 copies/mL)	0.37 (0.15–0.93)	0.035
**Toilet/washroom facilities**	Age: 20–29 years	0.46 (0.21–0.99)	0.048
	Age: 50–59 years	0.55 (0.31–0.98)	0.043
	Level of Education:Tertiary Education	0.41 (0.24–0.69)	0.008
**Cleanliness of the building**	Employment Status:Retired	0.22 (0.10–0.48)	<0.001

* Multivariable models adjusted for age, sex, ethnicity, marital status, education level, employment status, religion, nationality, and viral load. ^†^ Reference categories were selected based on the highest frequency of responses for each variable and are as follows: age: 40–49 years; level of education: secondary education; viral load: suppressed (<1000 copies/mL); employment status: employed.

## Data Availability

All data analyzed or generated during this study are included in the manuscript.

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
