# Peer review of "A Cross-Sectional Client Satisfaction Study Among Persons Living with HIV Attending a Large HIV Treatment Centre in Trinidad"

_healthcare, 2025, doi:10.3390/healthcare13121400_

Round 1
Reviewer 1 Report
Comments and Suggestions for Authors
The paper presents an interesting and relevant topic. Overall, the study is interesting, and the research problem is well-defined, with the main objective being to assess client satisfaction at an HIV clinic in Trinidad, focusing on identifying service gaps and factors associated with reduced viral suppression. However, the paper suffers from several drawbacks that need to be corrected before its possible acceptance.
- The abstract follows a logical structure (Background, Objectives, Methods, Results, Conclusion), which facilitates readability. But need more background to understand the problem.
- I advise the authors to professionally proofread their manuscript prior to resubmitting.
- Introduction is very very short. Need more literature review for undestanding the theme.
- Methods chapter needs more specification of empirical work. Although the use of a structured questionnaire is mentioned, details regarding its validation, the satisfaction measurement scale, and whether ethical consent was obtained are missing. Even in an abstract, a brief mention of these aspects would strengthen the study's credibility. While the use of multivariable logistic regression is a suitable methodological choice for identifying factors associated with satisfaction, this choice could be better justified.
- The results are focused and detailed. However, the conclusion could be more action-oriented — for example, suggesting interventions to improve wait times or physical facilities, rather than simply stating that they should be 'prioritized.
Author Response
Reviewer #1
Comments and Suggestions for Authors
The paper presents an interesting and relevant topic. Overall, the study is interesting, and the research problem is well-defined, with the main objective being to assess client satisfaction at an HIV clinic in Trinidad, focusing on identifying service gaps and factors associated with reduced viral suppression. However, the paper suffers from several drawbacks that need to be corrected before its possible acceptance.
- The abstract follows a logical structure (Background, Objectives, Methods, Results, Conclusion), which facilitates readability. But need more background to understand the problem.
Answer
Client satisfaction with HIV service delivery reflects the ability of healthcare providers to effectively deliver care and treatment that meets the requirements and expectations of clients and is associated with improved health outcomes including increased retention in care and HIV viral suppression
- I advise the authors to professionally proofread their manuscript prior to resubmitting.
Answer
The manuscript has been professionally proofread
- Introduction is very very short. Need more literature review for undestanding the theme.
Answer
The Introduction has been enhanced
Client satisfaction with HIV service delivery is associated with the perceived standard of care [7] and it reflects the ability of healthcare providers to effectively deliver care and treatment that meets the requirements and expectations of clients (Adissu et al, 2020). Patient satisfaction may be evaluated from the characteristics of the provider-patient relationship including the adequacy of staffing, privacy, confidentiality and communication with health care workers; the services offered and wait time at the clinic, and the health facility environment including location, accessibility, physical amenities, sanitation and hygiene (Adissu et al, 2020). Studies conducted in Cameroon (Wung et al, 2016), Ethiopia (Adissu et al, 2020, Niguisse et al 2020, Abdissa et al, 2024) and Nigeria (Okafoagu et al, 2013) have shown that patient satisfaction may be influenced by sociodemographic factors including age, sex, educational level, occupation and marital status (Adisssu et al, 2020, Niguisse et al, 2020). Client dissatisfaction with the clinical services may result from ineffective communication, long wait times, disrespect and perceived rudeness by the health care providers and poor health literacy (Strzelecka et al, 2020). Clients who are dissatisfied with the health services are less likely to adhere to their treatment regimens, have reduced service utilization and increased clinic default, may experience worse clinical outcomes and may have a tendency to get into potential disagreements with their providers (Strzelecka et al, 2021).
- Methods chapter needs more specification of empirical work. Although the use of a structured questionnaire is mentioned, details regarding its validation, the satisfaction measurement scale, and whether ethical consent was obtained are missing. Even in an abstract, a brief mention of these aspects would strengthen the study's credibility.
Answer
The questionnaire was adapted from a validated instrument used in the study by Buluba et al, 2021 (24 ) where the reliability of the tool was tested and the Cronbach’s alpha was 0.71. The questionnaire was pre-tested on 10 clients attending the HIV Clinic to identify and address problems/ambiguities before the main study was started. No formal validation of the questionnaire used in our study was done which may present psychometric weaknesses and reduce the possibility of more accurately interpreting the results related to patient satisfaction which is a limitation of the study. This limitation of the study which has been addressed in the discussion.
Ethical approval to conduct the study was obtained from the Campus Research Ethics Committee of the University of the West Indies, St Augustine approval number CREC-SA.2073/03/2023.
While the use of multivariable logistic regression is a suitable methodological choice for identifying factors associated with satisfaction, this choice could be better justified.
Answer
Multivariable logistic regression is a suitable methodological choice for analysing patient satisfaction/dissatisfaction based on multiple independent variables such as quality of service, sex, viral suppression, age groups, client wait time etc. This statistical methodology is particularly valuable because it can be used to distinguish binary outcomes (satisfaction/dissatisfaction) and complex associations among variables.
- The results are focused and detailed. However, the conclusion could be more action-oriented — for example, suggesting interventions to improve wait times or physical facilities, rather than simply stating that they should be 'prioritized.
Answer
The study showed that 93.1% of the study participants were virally suppressed, there was a general level of satisfaction with the overall care at the clinic, with the health care worker services and that privacy/confidentiality was maintained. Interventions such as infrastructure enhancements, efficient scheduling of patients and streamlining workflows to reduce wait time, educational programs focused on treatment adherence, and strategies that strengthen continuity of care will be implemented to improve service delivery especially in patients with a higher risk of defaulting from care and those with unsuppressed viral loads. Future longitudinal studies will be implemented to evaluate the effectiveness of these interventions using more advanced methodological approaches to strengthen public health policies.
Reviewer 2 Report
Comments and Suggestions for Authors
Thank you for the opportunity to review this manuscript. Below, my comments and suggestions, which I hope will help strengthen the manuscript and enhance its impact:
The introduction provides a good overview of the HIV pandemic globally and in Trinidad and Tobago. However, it could benefit from a more detailed discussion of prior studies on client satisfaction within HIV clinics, especially in comparable settings. A broader review of similar studies in low- and middle-income countries would enhance the background. Additionally, some of the references are dated and could be supplemented with more recent literature to reflect advancements in this area.
While the methods are described in reasonable detail, more information on the validation process for the questionnaire would be helpful. For example, details on how the questionnaire was pre-tested and whether it was adapted from previously validated instruments would strengthen the methodological rigor. Additionally, a brief description of how convenience sampling might affect generalizability should be included.
Several elements in the discussion could be strengthened to enhance the depth and relevance of the analysis. The connection between client satisfaction and HIV viral suppression (Lines 246–251) could be more thoroughly examined, particularly by analyzing how specific shortcomings—such as prolonged waiting times and inadequate physical infrastructure—may directly affect patient retention and adherence to therapy.
A more critical comparison with studies conducted in other regions (Lines 272–282) would also enrich the discussion, especially if the authors emphasize distinctive findings from this study, such as the influence of tertiary education on satisfaction levels. Similarly, the challenges related to the clinic’s visible location (Lines 287–294) merit further elaboration, including potential strategies to mitigate stigma, such as community outreach initiatives or the integration of telehealth services.
Limitations: while the manuscript appropriately acknowledges the use of convenience sampling and the exclusion of newly enrolled patients, additional considerations could be incorporated. Specifically, the potential for self-report bias (Lines 370–377) should be included, as participants may have overstated their satisfaction due to concerns about stigma or perceived repercussions.
Finally, the discussion would benefit from a stronger emphasis on the need for longitudinal studies (Lines 377–382) to assess how enhancements in service delivery might influence client satisfaction, treatment adherence, and HIV viral suppression over time. Taking these points into account would meaningfully improve the manuscript’s impact and usefulness in practice.
English: Revise sentences for clarity and conciseness. For example, breaking down long sentences and simplifying complex phrasing would improve readability. Ensure consistent use of technical terms and abbreviations throughout the manuscript.
The references cited are relevant to the research. However, more recent literature could be included to strengthen the article.
Author Response
Reviewer #2
Thank you for the opportunity to review this manuscript. Below, my comments and suggestions, which I hope will help strengthen the manuscript and enhance its impact:
The introduction provides a good overview of the HIV pandemic globally and in Trinidad and Tobago. However, it could benefit from a more detailed discussion of prior studies on client satisfaction within HIV clinics, especially in comparable settings. A broader review of similar studies in low- and middle-income countries would enhance the background.
Answer
Client satisfaction with HIV service delivery is associated with the perceived standard of care [7] and it reflects the ability of healthcare providers to effectively deliver care and treatment that meets the requirements and expectations of clients (Adissu et al, 2020). Patient satisfaction may be evaluated from the characteristics of the provider-patient relationship including the adequacy of staffing, privacy, confidentiality and communication with health care workers; the services offered and wait time at the clinic, and the health facility environment including location, accessibility, physical amenities, sanitation and hygiene (Adissu et al, 2020). Studies conducted in Cameroon (Wung et al, 2016), Ethiopia (Adissu et al, 2020, Niguisse et al 2020, Abdissa et al, 2024) and Nigeria (Okafoagu et al, 2013) have shown that patient satisfaction may be influenced by sociodemographic factors including age, sex, educational level, occupation and marital status (Adisssu et al, 2020, Niguisse et al, 2020). Client dissatisfaction with the clinical services may result from ineffective communication, long wait times, disrespect and perceived rudeness by the health care providers and poor health literacy (Strzelecka et al, 2020). Clients who are dissatisfied with the health services are less likely to adhere to their treatment regimens, have reduced service utilization and increased clinic default, may experience worse clinical outcomes and may have a tendency to get into potential disagreements with their providers (Strzelecka et al, 2021).
Additionally, some of the references are dated and could be supplemented with more recent literature to reflect advancements in this area.
Answer
References with more recent literature to reflect advancements in the area have been added
While the methods are described in reasonable detail, more information on the validation process for the questionnaire would be helpful. For example, details on how the questionnaire was pre-tested and whether it was adapted from previously validated instruments would strengthen the methodological rigor. Additionally, a brief description of how convenience sampling might affect generalizability should be included.
Answer
The questionnaire was adapted from a validated instrument used in the study by Buluba et al ( ) where the reliability of the tool was tested and the Cronbach’s alpha was 0.71. The questionnaire was pre-tested on 10 clients attending the HIV Clinic to identify and address problems/ambiguities before the main study was started. No formal validation of the questionnaire used in our study was done which may present psychometric weaknesses and reduce the possibility of more accurately interpreting the results related to patient satisfaction which is a limitation of the study. This was a cross-sectional study with convenience sampling which may result in bias and thus may not be generalizable to the other PLHIV attending the clinic as participants in the study were chosen on easy accessibility rather than by a random sampling method.
Several elements in the discussion could be strengthened to enhance the depth and relevance of the analysis. The connection between client satisfaction and HIV viral suppression (Lines 246–251) could be more thoroughly examined, particularly by analyzing how specific shortcomings—such as prolonged waiting times and inadequate physical infrastructure—may directly affect patient retention and adherence to therapy.
Answer
Increased wait times at HIV clinics may adversely impact HIV viral suppression by reducing patient engagement in care (as they have to take time off from work with the potential for loss of wages), medication adherence, and retention in care (Hickey et al, 2020, Simelane et al, 2022 ). A study by Hicket et al (37) showed that shorter wait times, can improve viral suppression, especially among those who are ART-experienced (Hickey et al, 2020). However, a study by De Schacht et al, 2024 39) showed that Increased time spent at a HIV treatment facility was not associated with a lower odds of viral suppression nor a lower retention in care (De Schacht et al, 2024).
In our study, 347 (95.9%) clients stated that their privacy/confidentiality was always maintained and similar results were obtained in a study by Buluba et al , 2021 in Tanzania [24], however this was in contrast to a study conducted by Tran et al in Vietnam, 2012 [32] where 60.1% of clients were satisfied with privacy and confidentiality of the health care services. In addition, poor physical infrastructure that result in overcrowding of the clinic and lack of privacy may impact patient engagement and retention in care (Mwamba et al, 2018)
A more critical comparison with studies conducted in other regions (Lines 272–282) would also enrich the discussion, especially if the authors emphasize distinctive findings from this study, such as the influence of tertiary education on satisfaction levels.
Answer
In our study, participants were less likely to be satisfied with the amount of medication received if they obtained a tertiary education, OR 0.44, 95% CI (0.21 – 0.94), p = 0.034, this is similar to studies conducted in four PEPFAR supported African countries (Somi et al, 2021), Ethiopia (Abebe et al, 2016) and Nigeria (Agu et al, 2014) which reported lower levels of satisfaction among those with higher education levels. Patients who are more educated may be better informed and thus have higher expectations of the health care services than those who are less educated and possibly less informed (Agu et al, 2014). In contrast, other studies conducted in Ethiopia reported higher levels of satisfaction among patients with higher levels of education compared to those with no formal education (Nigussie et al, 2020, Abdissa et al, 2024)
Similarly, the challenges related to the clinic’s visible location (Lines 287–294) merit further elaboration, including potential strategies to mitigate stigma, such as community outreach initiatives or the integration of telehealth services.
Answer
Due to stigma and discrimination, 29.9% of study participants expressed dissatisfaction with the location of the building as clients do not like to be seen entering the HIV clinic which is located on a busy thoroughfare and members of the key population groups such as MSM, transgender persons and sex workers who have a higher HIV prevalence experience additional stigma and discrimination due to their sexual identities and behaviours [36]. Therefore, clients are encouraged to wear masks (as COVID-19 protocols are still in place) or they are offered the option of attending the extended hours clinic from 3pm-7pm when fewer persons are around (Edwards et al 2021) or they can attend the two other MRFTT Clinic sites that are in more discrete locations. Other strategies to mitigate stigma include community outreach initiatives and the integration of telehealth services (which is already being done).
Limitations: while the manuscript appropriately acknowledges the use of convenience sampling and the exclusion of newly enrolled patients, additional considerations could be incorporated. Specifically, the potential for self-report bias (Lines 370–377) should be included, as participants may have overstated their satisfaction due to concerns about stigma or perceived repercussions.
Answer
The potential for self-report bias may be due to social desirability, as participants may exaggerate their satisfaction with the clinical services due to concerns about stigma or perceived ramifications. This has been included in the limitations of the study
Finally, the discussion would benefit from a stronger emphasis on the need for longitudinal studies (Lines 377–382) to assess how enhancements in service delivery might influence client satisfaction, treatment adherence, and HIV viral suppression over time. Taking these points into account would meaningfully improve the manuscript’s impact and usefulness in practice.
Answer
Interventions such as infrastructure enhancements, efficient scheduling of patients and streamlining workflows to reduce wait time, educational programs focused on treatment adherence, and strategies that strengthen continuity of care will be implemented to improve service delivery especially in patients with a higher risk of defaulting from care and those with unsuppressed viral loads. Future longitudinal studies will be implemented to evaluate the effectiveness of these interventions using more advanced methodological approaches to strengthen public health policies.
English: Revise sentences for clarity and conciseness. For example, breaking down long sentences and simplifying complex phrasing would improve readability. Ensure consistent use of technical terms and abbreviations throughout the manuscript.
Answer
The English has been revised
The references cited are relevant to the research. However, more recent literature could be included to strengthen the article.
Answer
More recent references have been added
Reviewer 3 Report
Comments and Suggestions for Authors
Thank you for the opportunity to review the manuscript “A Cross-Sectional Client Satisfaction Study Among Persons Living with HIV (PLHIV) Attending a Large HIV Treatment Centre in Trinidad”.
Overall, this study is valuable and deserves to be published as it provides relevant information on patient satisfaction in a large sample of PLHIV attending a major clinic in Trinidad. The study is well designed and has appropriate ethical approval. It addresses key aspects of health care, such as confidentiality, staff interaction, physical infrastructure, and viral suppression outcomes. The discussion is comprehensive and is supported by a current and diverse literature review. However, there are aspects of methodology that need to be substantially addressed before possible publication.
As an area for improvement, the authors should recognize that the questionnaire used presents psychometric weaknesses, since no formal validation is reported and no previous studies are cited to support its reliability and validity. This limits the possibility of more accurately interpreting the results related to patient satisfaction.
In addition, the predominant use of categorical variables (e.g., yes/no levels) restricts the application of more robust statistical analyses, such as correlations, regressions, or mediation models, which would have allowed a more in-depth exploration of factors associated with satisfaction and viral suppression. It is suggested that future studies use continuous measures and validated scales to enrich quantitative analyses and obtain more detailed conclusions or at least make mention of it in the study limitations.
Furthermore, although the sample size is adequate, more details on the recruitment of participants could be included to ensure the representativeness of the data and avoid selection bias.
Finally, based on the findings presented, it is recommended to implement interventions aimed at improving the aspects identified as problematic by the participants (such as physical facilities, waiting time and access to medication), especially in populations with a higher risk of treatment abandonment or lower viral suppression. These interventions could include infrastructure improvements, educational programs focused on treatment adherence, and strategies that strengthen continuity of care. Future studies could evaluate the effectiveness of such measures using more advanced methodological approaches to strengthen public health policies.
Author Response
Reviewer #3
Thank you for the opportunity to review the manuscript “A Cross-Sectional Client Satisfaction Study Among Persons Living with HIV (PLHIV) Attending a Large HIV Treatment Centre in Trinidad”.
Overall, this study is valuable and deserves to be published as it provides relevant information on patient satisfaction in a large sample of PLHIV attending a major clinic in Trinidad. The study is well designed and has appropriate ethical approval. It addresses key aspects of health care, such as confidentiality, staff interaction, physical infrastructure, and viral suppression outcomes. The discussion is comprehensive and is supported by a current and diverse literature review. However, there are aspects of methodology that need to be substantially addressed before possible publication.
As an area for improvement, the authors should recognize that the questionnaire used presents psychometric weaknesses, since no formal validation is reported and no previous studies are cited to support its reliability and validity. This limits the possibility of more accurately interpreting the results related to patient satisfaction.
Answer
The questionnaire was adapted from a validated instrument used in the study by Buluba et al, 2021 (24 ) where the reliability of the tool was tested and the Cronbach’s alpha was 0.71. The questionnaire was pre-tested on 10 clients attending the HIV Clinic to identify and addressed problems/ambiguities before the main study was started. No formal validation of the questionnaire used in our study was done which may present psychometric weaknesses and reduce the possibility of more accurately interpreting the results related to patient satisfaction which is a limitation of the study. This limitation of the study which has been addressed in the discussion.
In addition, the predominant use of categorical variables (e.g., yes/no levels) restricts the application of more robust statistical analyses, such as correlations, regressions, or mediation models, which would have allowed a more in-depth exploration of factors associated with satisfaction and viral suppression. It is suggested that future studies use continuous measures and validated scales to enrich quantitative analyses and obtain more detailed conclusions or at least make mention of it in the study limitations.
Answer
Thank you for the recommendations. The limitation of the use of categorical variables would be addressed in the discussion/limitations of the study.
The predominant use of categorical variables limited the application of more robust statistical analyses, such as correlations or regressions, which would have allowed a more detailed investigation of factors associated with satisfaction and viral suppression. It is recommended that future studies use continuous variables and validated scales to improve quantitative analyses so that more detailed conclusions can be obtained
Furthermore, although the sample size is adequate, more details on the recruitment of participants could be included to ensure the representativeness of the data and avoid selection bias.
Answer
Over the period April 15, 2023- March 31, 2024, participants were recruited for the study using convenience sampling. Approximately 80-120 clients attend the MRFTT HIV Clinic daily and the study was conducted on about 1-3 days per week depending on the availability of study staff. Patients were informed about the research study by a doctor or a nurse and asked about their willingness to participate. After obtaining written informed consent, the self-questionnaire was given to study participants. An average of 5-10 completed questionnaires were collected on each day the study was conducted.
Finally, based on the findings presented, it is recommended to implement interventions aimed at improving the aspects identified as problematic by the participants (such as physical facilities, waiting time and access to medication), especially in populations with a higher risk of treatment abandonment or lower viral suppression. These interventions could include infrastructure improvements, educational programs focused on treatment adherence, and strategies that strengthen continuity of care. Future studies could evaluate the effectiveness of such measures using more advanced methodological approaches to strengthen public health policies.
Answer
Interventions such as infrastructure enhancements, efficient scheduling of patients and streamlining workflows to reduce wait time, more efficient forecasting of medication to reduce the possibility of stock-outs, educational programs focused on treatment adherence, and strategies that strengthen continuity of care will be implemented to improve service delivery especially in patients with a higher risk of defaulting from care and those with lower viral suppression. Future longitudinal studies will be implemented to evaluate the effectiveness of these interventions using more advanced methodological approaches to strengthen public health policies.
Reviewer 4 Report
Comments and Suggestions for Authors
Introduction
The introduction should include a clear justification for the study. It is important to highlight the relevance of the research and what new insights it contributes to the scientific community.
Methodology
- The use of convenience sampling should be better justified, and its associated limitations discussed.
- It should be stated whether the instruments used were previously validated. What were the internal consistency values reported in earlier studies?
- The "Methods" section could benefit from the addition of subheadings, such as: "Population and Sample", "Procedures", and "Ethical Considerations".
Conclusion
- The practical implications of the study should be clearly stated (e.g., for future actions or health policy).
- Suggestions for future studies or research should also be included.
References
The reference list should be reviewed according to the journal’s formatting guidelines. For example, verify the formatting of the references in lines 420, 433, and 472.
Author Response
Reviewer #4
Introduction
The introduction should include a clear justification for the study. It is important to highlight the relevance of the research and what new insights it contributes to the scientific community.
Answer
Data are sparse regarding client satisfaction of the HIV treatment services in the Caribbean, thus it is important to assess these services from the patients’ perspective in order to understand the baseline, identify areas of concern and assess interventions in health care provision to the identified problems with the goal of achieving HIV viral suppression. The aim of the study is to conduct a client satisfaction survey among PLHIV attending the HIV Clinic, MRFTT to identify gaps in service delivery at the clinic and recognize factors that may contribute to reduced HIV viral suppression so that appropriate interventions can be put in place.
Methodology
- The use of convenience sampling should be better justified, and its associated limitations discussed.
Answer
This was a cross-sectional study with convenience sampling which may result in bias and thus may not be generalizable to the other PLHIV attending the clinic as participants in the study were chosen on easy accessibility rather than by a random sampling method.
- It should be stated whether the instruments used were previously validated. What were the internal consistency values reported in earlier studies?
Answer
The questionnaire was adapted from a validated instrument used in the study by Buluba et al, 2021 (24 ) where the reliability of the tool was tested and the Cronbach’s alpha was 0.71. The questionnaire was pre-tested on 10 clients attending the HIV Clinic to identify and address problems/ambiguities before the main study was started. No formal validation of the questionnaire used in our study was done which may present psychometric weaknesses and reduce the possibility of more accurately interpreting the results related to patient satisfaction which is a limitation of the study. This limitation of the study which has been addressed in the discussion.
- The "Methods" section could benefit from the addition of subheadings, such as: "Population and Sample", "Procedures", and "Ethical Considerations".
Answer
This has been completed in the manuscript as suggested
Conclusion
- The practical implications of the study should be clearly stated (e.g., for future actions or health policy).
- Suggestions for future studies or research should also be included.
Answer
Interventions such as infrastructure enhancements, efficient scheduling of patients and streamlining workflows to reduce wait time, more efficient forecasting of medication to reduce the possibility of stock-outs, educational programs focused on treatment adherence, and strategies that strengthen continuity of care will be implemented to improve service delivery especially in patients with a higher risk of defaulting from care and those with lower viral suppression. Future longitudinal studies will be implemented to evaluate the effectiveness of these interventions using more advanced methodological approaches to strengthen public health policies.
References
The reference list should be reviewed according to the journal’s formatting guidelines. For example, verify the formatting of the references in lines 420, 433, and 472.
Answer
The references have been reviewed according to the journal’s formatting guidelines
Round 2
Reviewer 2 Report
Comments and Suggestions for Authors
Thank you again for considering me as a reviewer.
I think that significant improvements have been made to the article, but I would still suggest some minor adjustments. Regarding the references, these could be enhanced. For instance, in relation to the use of dolutegravir, which is a standard medication in other countries, it would be advisable to include a citation from the last five years reflecting these differences with Trinidad and Tobago. Another improvement would be to include references and discussions on the impact of telemedicine and its effectiveness in managing patients with HIV.
I would recommend including a clarification about the questionnaire used, which has not been validated in the context of Trinidad and Tobago. While this might have limited the accuracy and validity of the results obtained, it would be advisable for future studies to conduct a formal validation process using psychometric analyses, among other methods. I am unaware if the research team has already considered this issue and prefers not to allude to ongoing research, but even then, I would include a paragraph reflecting that this matter is being taken into account.
Author Response
Thank you for these very insightful comments. Changes are highlighted green in the text of the manuscript
Response to Reviewers
Reviewer #2
Thank you again for considering me as a reviewer.
I think that significant improvements have been made to the article, but I would still suggest some minor adjustments. Regarding the references, these could be enhanced. For instance, in relation to the use of dolutegravir, which is a standard medication in other countries, it would be advisable to include a citation from the last five years reflecting these differences with Trinidad and Tobago.
Answer
Trinidad and Tobago is considered a high-income country and thus does not have access to generic dolutegravir due to patency issues [44], though tenofovir/lamivudine/dolutegravir (TLD) is used as a first line ART regimen in other Caribbean countries (Cushnie et al 2024)
Another improvement would be to include references and discussions on the impact of telemedicine and its effectiveness in managing patients with HIV.
Answer
Some benefits of telehealth include increased retention in care for patients who live long distances from clinic, privacy for patients who do not want to be seen attending an HIV clinic and greater flexibility in booking appointments (Smith et al 2021). Limitations of telehealth include disparities in the use of technology by racial minority groups, older individuals and those with low telehealth literacy and lack of access to appropriate devices, broadband internet and appropriate policies for patients’ data protection (labisi et al 2022).
I would recommend including a clarification about the questionnaire used, which has not been validated in the context of Trinidad and Tobago. While this might have limited the accuracy and validity of the results obtained, it would be advisable for future studies to conduct a formal validation process using psychometric analyses, among other methods. I am unaware if the research team has already considered this issue and prefers not to allude to ongoing research, but even then, I would include a paragraph reflecting that this matter is being taken into account.
Answer
It is recommended that future studies use continuous variables and a formal validation process using psychometric analyses to improve quantitative evaluations so that more detailed conclusions can be obtained
Reviewer 3 Report
Comments and Suggestions for Authors
The authors have addressed all the comments in detail.
Author Response
Thank you for all your insightful comments